

# Predictors of treatment failure during the first year in newly diagnosed type 2 diabetes patients: a retrospective, observational study

Hon-Ke Sia[1,2], Chew-Teng Kor[3], Shih-Te Tu[1], Pei-Yung Liao[1] and Yu-Chia Chang[2,4]

[1] Division of Endocrinology and Metabolism, Department of Internal Medicine, Changhua Christian Hospital, Changhua City, Taiwan
[2] Department of Healthcare Administration, Asia University, Taichung City, Taiwan
[3] Internal Medicine Research Center, Changhua Christian Hospital, Changhua City, Taiwan
[4] Department of Medical Research, China Medical University Hospital, China Medical University, Taichung City, Taiwan

## ABSTRACT

**Background**. Diabetes patients who fail to achieve early glycemic control may increase the future risk of complications and mortality. The aim of the study was to identify factors that predict treatment failure (TF) during the first year in adults with newly diagnosed type 2 diabetes mellitus (T2DM).

**Methods**. This retrospective cohort study conducted at a medical center in Taiwan enrolled 4,282 eligible patients with newly diagnosed T2DM between 2002 and 2017. Data were collected from electronic medical records. TF was defined as the HbA1c value >7% at the end of 1-year observation. A subgroup analysis of 2,392 patients with baseline HbA1c ≥8% was performed. Multivariable logistic regression analysis using backward elimination was applied to establish prediction models.

**Results**. Of all study participants, 1,439 (33.6%) were classified as TF during the first year. For every 1% increase in baseline HbA1c, the risk of TF was 1.17 (95% CI 1.15–1.20) times higher. Patients with baseline HbA1c ≥8% had a higher rate of TF than those with HbA1c <8% (42.0 vs 23.0%, $p < 0.001$). Medication adherence, self-monitoring of blood glucose (SMBG), regular exercise, gender (men), non-insulin treatment, and enrollment during 2010–2017 predicted a significant lower risk of TF in both of the primary and subgroup models.

**Conclusions**. Newly diagnosed diabetes patients with baseline HbA1c ≥8% did have a much higher rate of TF during the first year. Subgroup analysis for them highlights the important predictors of TF, including medication adherence, performing SMBG, regular exercise, and gender, in achieving glycemic control.

Corresponding author
Yu-Chia Chang, ycchang@asia.edu.tw

## INTRODUCTION

Diabetes mellitus (DM) is among the most serious chronic diseases worldwide. The prevention and treatment of diabetes is a major health care issue due to its high prevalence, related comorbidities, complications, and high related medical cost. Early glycemic control may have long-lasting (at least 10 years) effects in reducing the risk of severe microvascular and macrovascular complications, known as the legacy effect (metabolic memory) (*Chalmers & Cooper, 2008*; *Holman et al., 2008*). *Walraven et al. (2015)* reported that patients who responded quickly to glycemic control showed a lower prevalence of retinopathy and microalbuminuria. A large cohort study of newly diagnosed diabetes patients with at least 10-year survival showed that poor control (mean HbA1c $\geq$8.0%) during the first year was associated with increased future risk of microvascular events and mortality (*Laiteerapong et al., 2019*). These findings highlight the urgency of improving glycemic control in newly diagnosed diabetes patients.

Despite a tendency for better islet function in newly diagnosed patients with type 2 DM (T2DM), many still fail to achieve early glycemic control. A nationwide prospective cohort study reported that 31.5% of newly diagnosed Chinese diabetes patients failed to achieve HbA1c target levels (<7.0%) after 12 months of treatment (*Cai et al., 2019*). Early detection of the factors that predispose to treatment failure could help identify those at risk of not achieving glycemic control and enable tailoring of treatment measures.

Previous studies investigating predictors of poor glycemic control rarely focused on newly diagnosed T2DM patients (*Cai et al., 2019*; *Svensson et al., 2016*). There exist characteristic differences between newly diagnosed patients and those who had been on long-term treatment; thus, their predictors may also differ. *Ren et al. (2020)* reported that predictors of the response to anti-diabetic therapy differed between early- and advanced-stage T2DM. The findings of interventional studies may not reflect the situation in clinical practice, particularly medication adherence (*Blonde et al., 2018*; *Edelman & Polonsky, 2017*). Therefore, further studies focusing on newly diagnosed patients using real-world data are required to fill this information gap. The aim of the present study was to determine the major factors predicting treatment failure during the first year in adults with newly diagnosed T2DM.

## MATERIALS & METHODS

### Subjects

This retrospective cohort study was conducted at the Changhua Christian Hospital (CCH), Taiwan. A total of 24,473 patients with T2DM were enrolled in the Diabetes Care Management Program (DCMP) at the CCH Diabetes Care Center between January 2002 and December 2017. Patients were screened for eligibility using data from the hospital's electronic medical record system.

Patients diagnosed with T2DM, according to the criteria established by the American Diabetes Association, were included (*American Diabetes Association, 2019*). Those in whom the onset of diabetes occurred over 12 months prior to enrollment or at an age <30 years were excluded. The latter was to reduce the likelihood of type 1 diabetes. Patients aged

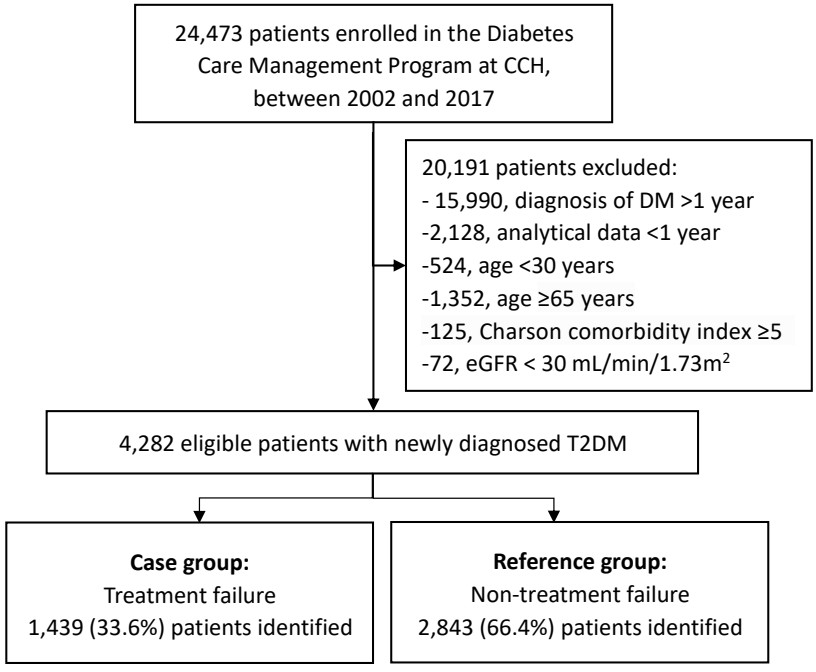

**Figure 1** **Flowchart of the study population.** Abbreviations: CCH, Changhua Christian Hospital; DM, diabetes mellitus; eGFR, estimated glomerular filtration rate.

≥65 years or with a Charlson comorbidity index scores ≥5 were excluded (*Charlson et al., 1987*; *Bannay et al., 2016*), considering that less stringent HbA1C goals (such as 8–8.5%) have been recommended for patients with limited life expectancy, extensive comorbid conditions, or frail, older adults since late 2000s (*American Diabetes Association, 2021*; *American Diabetes Association, 2021*). Patients with estimated glomerular filtration rate (eGFR) <30 mL/min/1.73 m² were also excluded as this may have affected the HbA1c level and not accurately reflect the true glycemic status (*Bloomgarden & Handelsman, 2018*). In the end, 4282 eligible patients with ≥ 1 year of analytical data were included (Fig. 1).

## Data collection

Data collected from the hospital's electronic medical record system included the DCMP diabetes registry, prescriptions, laboratory data, and CCH research database. Diabetes specialists referred patients with T2DM to the Diabetes Care Center to participate in the DCMP, usually 2 to 6 weeks after the first outpatient clinic visit. All patients received basic data registry, underwent health-related behavior survey, physical examination, and laboratory testing. They attended standardized one-to-one diabetes self-management (DSM) education classes upon enrollment into the DCMP. After completing the course, a certified diabetes educator conducted face-to-face interviews and evaluated and recorded each patient's frequency of performing self-monitoring of blood glucose (SMBG), knowledge regarding glycemic control, willingness toward DSM, and medication adherence.

## Outcome measurement

Treatment failure (TF) was defined as the HbA1c value >7% at the end of 1-year observation. The others with HbA1c levels ≤7% at the end point were categorized as non-TF (reference group). Serum HbA1c was measured through ion-exchange high-performance liquid chromatography using the VARIANTTM II Turbo system.

## Other variables

Basic data included age at onset of diabetes, gender, level of education, and family history of diabetes. Health-related behaviors included current smoking (tobacco use within the preceding year), drinking (alcohol consumption more than once per week within the preceding year), and physical activity [regular (≥30 min/day, ≥3 days/week), occasional (low level of exercise less than the regular exercise criteria) or no exercise]. SMBG was defined as self-assessment of blood glucose levels using a glucometer more than once per week. Knowledge regarding glycemic control was defined as an understanding of the need for and methods of controlling blood glucose. Willingness toward DSM was defined as the motivation to learn self-management techniques. Medication adherence was defined as taking medication regularly at the dose recommended by the physician over the past week. Four-point scales were used to assess the three aforementioned variables. Data were merged into simple dichotomies (i.e., top-two-box vs. bottom-two-box) and categorized as adequate (yes) or inadequate (no) for analysis.

Physical examination included measurement of blood pressure (BP), height, and body weight. Systolic BP and diastolic BP were measured with patients in a seated position after a 10-min rest. The mean BP was calculated as (1/3 SBP + 2/3 DBP). Body mass index (BMI) was calculated as body weight (kg)/height (m)$^2$. Baseline laboratory data, including total cholesterol (TC), high-density lipoprotein cholesterol (HDL-C), triglycerides (TG), low-density lipoprotein cholesterol (LDL-C), creatinine, and glutamic pyruvic transaminase (GPT) levels were measured using a UniCel DxC 800 Synchron Clinical System (Beckman Coulter, Brea, CA, USA). The eGFR was calculated using the equation recommended by the National Kidney Foundation (*Levey et al., 2003*).

Individual anti-diabetic medication use during the first six months was categorized as oral anti-diabetic drugs (OAD) alone, insulin alone, both, or none. Only medication used for >1 month was included. Data on the 19 major non-psychiatric comorbidities in the Charlson comorbidity index during the year preceding enrollment were collected for each patient from the CCH research database. Major comorbidities including congestive heart failure, coronary artery disease, and cerebrovascular accident were analyzed as independent variables. Enrollment time was classified into two categories: 2002–2009 and 2010–2017.

## Statistical analysis

Data were expressed as frequency with percentage and mean ± standard deviation for categorical and continuous covariates respectively. Univariable logistic regression analysis was performed to calculate odds ratios (ORs) of TF vs non-TF for all variables. Subsequently, multivariable logistic regression analysis was performed to establish prediction models adjusted for significant covariates as shown in Table 1. The backward stepwise regression

**Table 1  Basic characteristics of newly diagnosed type 2 diabetes patients: TF vs non-TF group.**

| | TF ($n = 1439$) | Non-TF ($n = 2843$) | OR (95% CI) | *p*-value |
|---|---|---|---|---|
| Age at onset (years) | 50.2 ± 8.5 | 51.3 ± 8.5 | 0.98 (0.98,0.99) | <0.001 |
| Gender: Men | 730 (50.7%) | 1631 (57.4%) | 0.77 (0.67,0.87) | <0.001 |
| Level of education: No | 112 (7.8%) | 130 (4.6%) | 1 | |
| Primary school | 478 (33.2%) | 803 (28.2%) | 0.69 (0.52,0.91) | 0.009 |
| High school | 654 (45.5%) | 1282 (45.1%) | 0.59 (0.45,0.78) | <0.001 |
| University or above | 195 (13.6%) | 628 (22.1%) | 0.36 (0.27,0.49) | <0.001 |
| Family history of DM: Yes | 709 (49.3%) | 1502 (52.8%) | 0.87 (0.76,0.98) | 0.028 |
| Current smoking | 310 (21.5%) | 492 (17.3%) | 1.31 (1.12,1.54) | <0.001 |
| Alcohol drinking | 95 (6.6%) | 233 (8.2%) | 0.79 (0.62,1.01) | 0.064 |
| Physical activity: No exercise | 899 (63.4%) | 1396 (49.5%) | 1 | |
| Occasional exercise | 220 (15.5%) | 526 (18.7%) | 0.65 (0.54,0.78) | <0.001 |
| Regular exercise | 300 (21.1%) | 896 (31.8%) | 0.52 (0.45,0.61) | <0.001 |
| Knowledge regarding GC: Yes | 710 (54.2%) | 1811 (68.7%) | 0.54 (0.47,0.62) | <0.001 |
| Willingness toward DSM: Yes | 1056 (80.6%) | 2220 (84.3%) | 0.77 (0.65,0.92) | 0.004 |
| Perform SMBG: Yes | 270 (18.8%) | 816 (28.7%) | 0.57 (0.49,0.67) | <0.001 |
| Medication adherence: Yes | 1353 (94.0%) | 2752 (96.8%) | 0.52 (0.38,0.70) | <0.001 |
| Clinical variables | | | | |
| HbA1c at baseline (%) | 9.8 ± 2.5 | 8.7 ± 2.6 | 1.17 (1.15,1.20) | <0.001 |
| HbA1c at 3-month (%) | 7.9 ± 1.6 | 6.5 ± 1.0 | 2.61 (2.42,2.81) | <0.001 |
| HbA1c at 6-month (%) | 7.9 ± 1.5 | 6.3 ± 0.8 | 4.91 (4.41,5.47) | <0.001 |
| HbA1c at 9-month (%) | 8.0 ± 1.5 | 6.3 ± 0.7 | 8.51 (7.42,9.76) | <0.001 |
| HbA1c at 12-month (%) | 8.3 ± 1.3 | 6.2 ± 0.5 | – | – |
| BMI (kg/m$^2$) | 26.7 ± 4.6 | 26.6 ± 4.3 | 1.00 (0.99,1.02) | 0.569 |
| Mean BP (mmHg) | 97.7 ± 12.6 | 96.4 ± 12.1 | 1.01 (1.00,1.01) | 0.001 |
| Total cholesterol (mg/dL) | 192.9 ± 47.1 | 181.0 ± 40.2 | 1.07 (1.05,1.08)[a] | <0.001 |
| Triglycerides (mg/dL) | 182.2 ± 193.9 | 152.2 ± 138.4 | 1.01 (1.01,1.02)[a] | <0.001 |
| HDL-C (mg/dL) | 47.7 ± 17.2 | 46.3 ± 11.9 | 1.01 (1.00,1.01) | 0.003 |
| LDL-C (mg/dL) | 113.7 ± 35.4 | 106.7 ± 33.0 | 1.06 (1.04,1.08)[a] | <0.001 |
| eGFR (mL/min/1.73m$^2$) | 98.5 ± 41.1 | 96.5 ± 30.7 | 1.02 (1.00,1.04)[a] | 0.076 |
| GPT (U/L) | 36.7 ± 31.8 | 34.9 ± 35.0 | 1.02 (1.00,1.03)[a] | 0.109 |
| Anti-diabetic medication | | | | |
| None or OAD alone | 1203 (83.6%) | 2550 (89.7%) | 1 | |
| Insulin alone | 63 (4.4%) | 61 (2.2%) | 2.19 (1.53,3.13) | <0.001 |
| OAD+ insulin | 173 (12.0%) | 232 (8.2%) | 1.58 (1.28,1.95) | <0.001 |
| Anti-hypertension agent | 670 (46.6%) | 1345 (47.3%) | 0.97 (0.85,1.10) | 0.643 |
| Use of statins | 786 (54.6%) | 1577 (55.5%) | 0.97 (0.85,1.10) | 0.598 |
| Use of fibrates | 215 (14.9%) | 338 (11.9%) | 1.30 (1.08,1.57) | 0.005 |
| Comorbidity: CCI score | 1.6 ± 0.9 | 1.6 ± 0.9 | 0.94 (0.88,1.02) | 0.123 |
| CHF | 119 (8.3%) | 279 (9.8%) | 0.83 (0.66,1.04) | 0.101 |
| CAD | 82 (5.7%) | 167 (5.9%) | 0.97 (0.74,1.27) | 0.817 |
| CVA | 57 (4.0%) | 100 (3.5%) | 1.13 (0.81,1.58) | 0.466 |

*(continued on next page)*

**Table 1** (*continued*)

| | TF (*n* = 1439) | Non-TF (*n* = 2843) | OR (95% CI) | *p*-value |
|---|---|---|---|---|
| Enrollment time: 2002-2009 | 959 (66.6%) | 1393 (49.0%) | 1 | |
| 2010–2017 | 480 (33.4%) | 1450 (51.0%) | 0.48 (0.42,0.55) | <0.001 |

**Notes.**

Results are expressed as mean ± SD or *n* (%).

[a]Odds ratio was calculated by per 10 units increase

TF, treatment failure; Non-TF, Non-treatment failure; SD, standard deviation; OR, odds ratio; CI, confidence interval; DM, diabetes mellitus; GC, glycemic control; HbA1c, hemoglobin A1c; BMI, body mass index; BP, blood pressure; HDL-C, high-density lipoprotein cholesterol; LDL-C, low-density lipoprotein cholesterol; eGFR, estimated glomerular filtration rate; GPT, glutamic pyruvic transaminase; OAD, oral anti-diabetic drug; CCI, Charlson comorbidity index; CHF, congestive heart failure; CAD, coronary artery disease; CVA, cerebrovascular accident.

was performed to be a variable selection method to avoid overfitting. We used tolerance and variance inflation factor (VIF) to detect whether there is multicollinearity between covariates in every model. If the value of VIF is less than 2, the multicollinearity problem is considered absent. Area under the receiver operating characteristic curve (AUC) and R-square were used to assess the predictive ability of the models for predicting TF. We performed a subgroup analysis of patients with baseline HbA1c ≥8% to demonstrate the effect of initial poor glycemic status on TF. All tests were two-tailed with a significance level of 0.05. IBM SPSS Statistics version 22 (IBM Corp., Armonk, NY, USA) was used for the analyses.

## Ethics statement

The study was approved by the Institutional Review Board of Changhua Christian Hospital (CCH IRB No: 191212). Informed consent was waived.

## RESULTS

We identified 4282 eligible patients (mean age, 50.9 ± 8.5 years; 55.1% men) between 2002 and 2017. Among these patients, 1439 (33.6%) were categorized as the TF group. Compared with the non-TF group, the TF group was younger (50.2 vs 51.3 years, $p < 0.001$) and included more current smokers (21.5% vs 17.3%, $p < 0.001$), whereas the distribution of BMI, alcohol drinking, eGFR, GPT, use of statins, and comorbidities were similar. Patients had lower levels of education, had no family history of diabetes, and women were predisposed to TF (Table 1).

Higher baseline HbA1c level, lipid levels (TC, HDL-C, LDL-C and TG), and mean BP indicated higher risk of TF. For every 1% the increase in baseline HbA1c, the risk of TF was 1.17 (95% CI [1.15–1.20]) times higher. Use of fibrates and insulin (alone or combined with OAD) during the first 6 months predicted greater TF. Enrollment during 2010–2017, regular exercise, good medication adherence, performing SMBG, good knowledge regarding glycemic control, and adequate willingness toward DSM reduced risk of TF.

According to baseline HbA1c level, the study subjects were divided into two subgroups. The higher HbA1c subgroup was composed of 2,392 patients with HbA1c ≥8%, including 1005 (42.0%) with TF. In contrast, only 434 (23.0%) of the 1,890 patients with HbA1c <8% had TF during the first year. Therefore, two prediction models were established: the primary model, which consisted of all study subjects, and the subgroup model, which consisted of a

**Table 2  Models to predict treatment failure by multivariable logistic regression analysis using backward elimination method.** Primary model: all study participants; Subgroup model: subgroup analysis of patients with baseline HbA1c ≥8%.

| | Primary model ($n = 4282$) | | Subgroup model ($n = 2392$) | |
|---|---|---|---|---|
| | OR (95% CI) | *p*-value | OR (95% CI) | *p*-value |
| HbA1c at baseline (%) | 1.17 (1.14,1.2) | <0.001 | | |
| Age at onset (years) | 0.98 (0.97,0.99) | <0.001 | | |
| Gender: Men | 0.64 (0.54,0.75) | <0.001 | 0.65 (0.54,0.78) | <0.001 |
| Level of education: No | 1 | | | |
|     Primary school | 0.92 (0.68,1.26) | 0.62 | | |
|     High school | 0.77 (0.56,1.07) | 0.12 | | |
|     University or above | 0.55 (0.38,0.8) | 0.002 | | |
| Current smoking | 1.39 (1.14,1.69) | 0.001 | | |
| Physical activity: No exercise | 1 | | 1 | |
|     Occasional exercise | 0.86 (0.71,1.04) | 0.13 | 0.91 (0.71,1.16) | 0.44 |
|     Regular exercise | 0.68 (0.57,0.81) | <0.001 | 0.59 (0.48,0.74) | <0.001 |
| Perform SMBG: Yes | 0.73 (0.61,0.89) | 0.002 | 0.63 (0.49,0.8) | <0.001 |
| Medication adherence: Yes | 0.53 (0.36,0.79) | 0.002 | 0.38 (0.23,0.66) | <0.001 |
| Mean BP (mmHg) | 1.01 (1.00,1.01) | 0.014 | | |
| Triglycerides (mg/dL) | 1.01 (1.00,1.01)[a] | 0.004 | 1.01 (1.00,1.01)[a] | 0.054 |
| Total cholesterol (mg/dL) | | | 1.03 (1.00,1.05)[a] | 0.031 |
| Anti-diabetic medication | | | | |
|     None or OAD alone | 1 | | 1 | |
|     Insulin alone | 2.15 (1.44,3.21) | <0.001 | 1.97 (1.24,3.15) | 0.004 |
|     OAD+ insulin | 1.35 (1.06,1.73) | 0.015 | 1.36 (1.03,1.79) | 0.031 |
| Enrollment time, 2010–2017 | 0.67 (0.57,0.79) | <0.001 | 0.6 (0.49,0.73) | <0.001 |
| R square | 0.136 | | 0.111 | |
| AUC for model | 0.694 | | 0.673 | |

**Notes.**
[a]Odds ratio was calculated by per 10-unit increase.

OR, odds ratio; CI, confidence interval; HbA1c, hemoglobin A1c; GC, glycemic control; SMBG, self-monitoring of blood glucose; BP, blood pressure; OAD, oral anti-diabetic drug; AUC, area under curve.

subgroup of patients with baseline HbA1c ≥8.0%, using multivariable backward stepwise logistic regression analysis (Table 2). Collinearity diagnostic tests showed an absence of multicollinearity between factors, where values for tolerance ranged from 0.597 to 0.993, corresponding to VIFs of 1.675 to 1.007 (Table S1). Men, regular exercise, performing SMBG, medication adherence and enrollment during 2010–2017 predicted a lower risk of TF in both models. Higher baseline HbA1c, younger age at onset, lower levels of education, and higher mean BP increased the risk of TF in the primary model, but the increase was not statistically significant in the subgroup model. Using insulin within the first 6 months was predictive of TF. Although high TG indicated a higher risk of TF in the primary model, it was replaced by high TC in the subgroup model.

## DISCUSSION

Previous studies on predictive factors or model of newly diagnosed T2DM were predominantly based on baseline HbA1c, which is a strong major predictor (*Walraven et al., 2015*; *Cai et al., 2019*; *Svensson et al., 2016*; *Laiteerapong et al., 2017*; *Hertroijs et al., 2018*). Higher baseline HbA1c may reflect poor beta cell function or prolonged hyperglycemia due to delayed diagnosis of DM (*Svensson et al., 2016*; *Laiteerapong et al., 2017*). Consistent with aforementioned studies, patients with baseline HbA1c ≥8% had a higher rate of TF than those with HbA1c <8% (42.0 vs 23.0%, $p < 0.001$). However, it is worth noting that baseline HbA1c became an insignificant predictor in the subgroup model after adjusting for other factors. In other words, further increase in baseline HbA1c ≥8% may raise a limited risk of TF. Other factors, including gender, SMBG, medication adherence, and regular exercise may be more predictive in newly diagnosed patients with baseline HbA1c ≥8%.

Medication non-adherence is common and may account for up to 75% of the gap in clinical efficacy between randomized controlled trial and real-world results in HbA1c reduction (*Nichols, Conner & Brown, 2010*; *Giugliano et al., 2018*). The present study showed that medication adherence is associated with a greater protection than other modifiable variables, especially in the subgroup model, indicating it may be more influential in reducing the risk of TF in patients with baseline HbA1c ≥8%. It supports clinicians to aggressively promote patients' medication adherence, especially those with high baseline HbA1c.

SMBG has been shown to improve glycemic control among diabetes patients using insulin, although its value for those with non-insulin-treated T2DM has remained inclusive (*Young et al., 2017*). Our study demonstrated that performing SMBG is associated with lower risk of TF in patients with newly diagnosed T2DM, which supports the International Diabetes Federation guideline recommending SMBG should be considered at the time of diagnosis for patients with T2DM as a part of their education (*International Diabetes Federation, 2020*).

Higher level of education was positively correlated with good medication adherence, SMBG, adequate knowledge regarding glycemic control, willingness toward DSM, and regular exercise (Table 3). Our findings are consistent with those of a previous study in Taiwan that showed that higher educational attainment was significantly associated with better understanding of health education and instructions, adequate health literacy, and better glycemic control (*Chen et al., 2014*). Knowledge regarding glycemic control and willingness toward DSM were not significant predictors in both of the primary and subgroup analyses, indicating that self-care behaviors (such as medication adherence, performing SMBG, and regular exercise) are more predictive of TF than knowledge or willingness in our models.

Smoking has been shown to increase diabetes incidence in general population (*Akter, Goto & Mizoue, 2017*). Additional, an adverse association between smoking and glycemic control has been reported (*Ohkuma et al., 2015*). Smoking may affect glucose homeostasis through several mechanisms, including increased insulin resistance, reduced insulin action,

**Table 3  Correlations between demographic variables and self-care factors for diabetes management.**

| Variables | 1 | 2 | 3 | 4 | 5 | 6 | 7 | 8 | 9 |
|---|---|---|---|---|---|---|---|---|---|
| 1. Education level | 1 | 0.26** | 0.059** | 0.057** | 0.096** | 0.19** | −0.44** | 0.26** | 0.055** |
| 2. Knowledge regarding GC | | 1 | 0.21** | 0.055** | 0.20** | 0.30** | −0.036* | 0.036* | −0.033* |
| 3. Willingness toward DSM | | | 1 | 0.036* | 0.042** | 0.079** | −0.011 | 0.01 | −0.020 |
| 4. Medication adherence | | | | 1 | 0.053** | 0.043** | −0.008 | 0.02 | −0.006 |
| 5. Physical activity | | | | | 1 | 0.16** | 0.14** | 0.019 | −0.092** |
| 6. Perform SMBG | | | | | | 1 | −0.061** | 0.053** | −0.016 |
| 7. Age (years) | | | | | | | 1 | −0.11** | −0.12** |
| 8. Gender (Men) | | | | | | | | 1 | 0.39** |
| 9. Current smoking | | | | | | | | | 1 |

Notes.

[*,**] Kendall's tau rank correlation coefficient was used. [*] $p < 0.05$, [**] $p < 0.01$.

GC, glycemic control; DSM, diabetes self-management; SMBG, self-monitoring of blood glucose.

and loss of islet $\beta$-cell mass (*Ohkuma et al., 2015*; *Śliwińska Mossoń & Milnerowicz, 2017*). Our finding highlights that smoking is associated with increased possibility of TF in patients with newly diagnosed T2DM. Therefore, taking action to the modifiable unhealthy behavior is essential for this population.

Our study showed that women are more likely to have TF. The finding is in line with a longitudinal study of 1450 Chinese with diabetes in Hong Kong that reported patients with unimproved control had a female preponderance (*Yin et al., 2016*). Another Germany multicenter study of 9108 patients with T2DM showed that men had greater HbA1c reduction than women after treatment (*Schütt et al., 2015*). By contrast, other studies have shown that women more likely to have glycemic control in comparison with men (*Casagrande et al., 2013*; *McCoy et al., 2017*). Gender differences in glycemic control may involve biological, psychosocial factors, health behaviors, responses to therapeutic interventions, self-diabetes management, and differential perception and impact of social support (*Schütt et al., 2015*; *Mondesir et al., 2016*). We suggest that gender difference should be considered in treating patients with newly diagnosed diabetes.

The present study showed that older age reduced the risk of TF in newly diagnosed T2DM patients, which was consistent with most previous studies (*Walraven et al., 2015*; *Laiteerapong et al., 2017*; *Nichols, Conner & Brown, 2010*; *Martono et al., 2015*). While older patients tended to have more unfavourable factors, such as less knowledge regarding glycemic control, less likely to perform SMBG and lower level of education, they had a lower risk of TF in the first year (Table 3). There remains a lack of consensus on the mechanisms underlying the inverse association of age and HbA1c. It may involve age-related differences in the pathogenesis of T2DM (*Martono et al., 2015*; *Chang & Halter, 2003*; *Geloneze et al., 2014*; *Scott et al., 2020*). Previous studies showed insulin therapy, either alone or combined with OAD, was associated with a higher risk of TF (*Cai et al., 2019*; *Benoit et al., 2005*). A common explanation is that insulin users have more severe beta cell loss and are therefore prone to TF.

The strengths of this study include its large sample size, the focus on newly diagnosed T2DM and further identification of predictors in patients with baseline HbA1c ≥8%. The

National Health Insurance in Taiwan covers more than 99% of the country's 23 million people and provides easy access to medical services (*Wu, Majeed & Kuo, 2010*). Therefore, the treatment and outcome in the study were less affected by insurance factors.

Our study had several limitations. First, the study participants were enrolled across a long-time window, and thus available first-line pharmacological treatment options varied considerably. We incorporated enrollment time as a variable in the regression models to reduce the confounding effect. Second, patients attending a medical center may have higher disease severity. Therefore, we adjusted relevant variables for comorbidity and performed subgroup analysis of patients with HbA1c $\geq 8\%$ to reduce the selection bias. Third, the occurrence of TF during the first-year may reflect either a specific medication intolerance, that often prevents patients maintaining their therapy, or therapeutic inertia, which is a non-negligible problem among diabetes care providers. Additionally, income, dietary habits, and occupation may also contribute to glycemic control. Our models were limited by the absence of related information.

Fourth, selection bias may exist since our study population did not include those patients with missing data or <1-year follow-up. Fifth, some of the data were self-reported, such as medication adherence and SMBG frequency, and social desirability bias could be a problem. Finally, the generalizability of the real-world study findings may be limited to settings with similar medical and sociocultural environment.

## CONCLUSIONS

Baseline HbA1c has been an important indicator in clinical treatment guidelines to assess the severity of glycemic control and guide clinicians to use initial OAD combination therapy or even insulin therapy (*American Diabetes Association, 2020*). The current study showed that patients with baseline HbA1c $\geq 8\%$ did have a much higher rate of TF. However, subgroup analysis for them demonstrated that when baseline HbA1c above 8%, the increase in HbA1c did not further raise the risk of TF. Other factors, including medication adherence, regular exercise, performing SMBG, using insulin, and gender became more predictive than baseline HbA1c level.

### Funding
This study was funded by grant 109-CCH-IRP-009 from the Changhua Christian Hospital Research Foundation. The funders had no role in study design, data collection and analysis, decision to publish, or preparation of the manuscript.

### Grant Disclosures
The following grant information was disclosed by the authors:
Changhua Christian Hospital Research Foundation: 109-CCH-IRP-009.

### Competing Interests
The authors declare there are no competing interests.

## Author Contributions

- Hon-Ke Sia conceived and designed the experiments, performed the experiments, analyzed the data, prepared figures and/or tables, authored or reviewed drafts of the paper, and approved the final draft.
- Chew-Teng Kor performed the experiments, analyzed the data, prepared figures and/or tables, authored or reviewed drafts of the paper, and approved the final draft.
- Shih-Te Tu and Pei-Yung Liao analyzed the data, authored or reviewed drafts of the paper, and approved the final draft.
- Yu-Chia Chang conceived and designed the experiments, analyzed the data, prepared figures and/or tables, authored or reviewed drafts of the paper, and approved the final draft.

## Human Ethics

The following information was supplied relating to ethical approvals (i.e., approving body and any reference numbers):

The study was approved by the Institutional Review Board of Changhua Christian Hospital, Taiwan (CCH IRB No: 191212).

## Data Availability

Data is available in the Supplemental Files.

## Supplemental Information

Supplemental information for this article can be found online at http://dx.doi.org/10.7717/peerj.11005#supplemental-information.

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
