# Peer review of "Predictors of treatment failure during the first year in newly diagnosed type 2 diabetes patients: a retrospective, observational study"

_PeerJ, doi:10.7717/peerj.11005_

## Round 0.1 · original submission · Major Revisions

Both reviewers 1 and 2 raise issues on the experimental design, and especially on the choice of the HbA1c threshold for patients' categorisation.

Reviewer 1 ·

Basic reporting

In this well-written, retrospective cohort study, Sia and colleagues investigated the clinical predictors of first-year treatment failure in 5579 Asian patients newly diagnosed with type 2 diabetes mellitus (T2DM). As demonstrated in the UKPDS trial, achieving early glycemic control can significantly reduce the long-term risk of microvascular and macrovascular complications in newly diagnosed T2DM patients, however, many uncertainties and controversies surround the possible predictors of poor glycemic responses in real life scenarios and further evidences are required in this field. Despite its interesting subject matter, unfortunately, the present investigation has major methodological flaws that would inflate the validity of the findings and predictive modeling.

Experimental design

1. The study participants were enrolled across a very long-time window (Jan 2002-Dec 2017, 16 years), thus implying that available first-line pharmacological treatment options (e.g. metformin, insulin secretagogues, DPP4 inhibitors, SGLT2 inhibitors, acarbose, TZDs, GLP1RAs, insulin) varied considerably, depending on whether these newly diagnosed T2DM patients initiated treatment earlier or later in the study window. Other than a different glycemic efficacy, the occurrence of first-year treatment failure may reflect either a specific medication intolerance, that often prevents patients maintaining their therapy, or therapeutic inertia, which is a non-negligible problem among diabetes care providers. Failing to control for these confounding variables, and their combined effects, may have distorted the associations, and should be regarded as an important study limitation.

2. Beside pharmacological treatment options, also glycemic targets have varied considerably during this wide study window, so that the definition of treatment failure used (“never achieving post-treatment HbA1c <8% at 3, 6, 9, or 12 months after initiating treatment during the first year”, lines 94-95) appears misleading. It is true that before 2004, the American Diabetes Association (ADA) recommended that HbA1c not be allowed to exceed 8.0%. However, since 2004, the ADA dropped the traditional 8.0% “action threshold” in favor of a general recommendation to treat most patients to <7.0%, especially those newly diagnosed with T2DM and expected to get clinical advantage of metabolic legacy. Few exceptions to this general recommendation are represented by frail, older adults (> 65 years of age), in which less stringent HbA1c targets are recommended (8-8.5%) since late 2000s. I would suggest narrowing the study window to ensure more homogeneous criteria for treatment and outcomes of the study participants, redefining treatment failure, and excluding elderly patients with high Charlson comorbidity index scores from the primary analysis.

3. Although legitimate, backward stepwise regression modeling is particularly subjected to multicollinearity problems when the predictive variables are intercorrelated. The methods section should explain how multicollinearity was handled.

Validity of the findings

See the points above.

Reviewer 2 ·

Basic reporting

No comment

Experimental design

No comment

Validity of the findings

No comment

Additional comments

This paper focuses on an interesting topic. However, there is a major concern that must be addressed.
The authors stated that “Participants with at least one of the four post-treatment HbA1c levels <8% were categorized as non-TF (reference group)”. This critical choice must be strongly supported, otherwise it appears arbitrary and invalidates the study conclusions. Why didn’t they choose only the HbA1c value at the end of the observation? In my opinion this value is more appropriate as cut-off between TF and non-TF. In any case, please, show in table the HbA1c values at the four assessments.

---

## Round 0.2 · accepted · Accept

All the issues raised by the reviewers have been satisfactorily addressed.

Reviewer 1 ·

Basic reporting

In their replies, the authors have satisfactorily addressed the major issues raised by this reviewer. The manuscript has been significantly improved and is now worth of publication. I do not have any further comments.

Experimental design

No comment

Validity of the findings

No comment

Reviewer 2 ·

Basic reporting

No comment

Experimental design

No comment

Validity of the findings

No comment

Additional comments

My concerns have been addressed, thus the paper is now susceptible for publication.

---

## Author Rebuttal · Round 0.2

# Responses to the Reviewers' Comments

Prof. Daniela Foti
Academic Editor
31 December 2020

Dear Dr. Daniela Foti,

Please find attached our revised manuscript titled "**Predictors of treatment failure during the first year in newly diagnosed type 2 diabetes patients: an observational study**", which was resubmitted for publication as an *Original Article* in *PeerJ*.

We appreciate the great effort made by the reviewer to improve the quality of our manuscript. In response to the reviewer, especially on the choice of the HbA1c threshold for patients' categorisation, we have revised our models with narrowing the study window and redefining the definition of outcome (treatment failure). Additionally, we have revised our manuscript according to the revised models. We sincerely believe that our revised manuscript becomes more clarified with satisfactory quality.

We look forward to hearing from you at your earliest convenience.

Sincerely,

Hon-Ke Sia, MD, MSc
e-mail: 90279@cch.org.tw
Division of Endocrinology and Metabolism, Department of Internal Medicine
Changhua Christian Hospital, 135 Nan-Hsiao Street, Changhua City 500, Taiwan

Yu-Chia Chang, PhD
e-mail: ycchang@asia.edu.tw
Department of Healthcare Administration, Asia University
500 Lioufong Rd., Wufeng, Taichung City 41354, Taiwan

## Response to Reviewer 1

**Basic reporting**

In this well-written, retrospective cohort study, Sia and colleagues investigated the clinical predictors of first-year treatment failure in 5579 Asian patients newly diagnosed with type 2 diabetes mellitus (T2DM). As demonstrated in the UKPDS trial, achieving early glycemic control can significantly reduce the long-term risk of microvascular and macrovascular complications in newly diagnosed T2DM patients, however, many uncertainties and controversies surround the possible predictors of poor glycemic responses in real life scenarios and further evidences are required in this field. Despite its interesting subject matter, unfortunately, the present investigation has major methodological flaws that would inflate the validity of the findings and predictive modeling.

**Experimental design**

1. The study participants were enrolled across a very long-time window (Jan 2002-Dec 2017, 16 years), thus implying that available first-line pharmacological treatment options (e.g. metformin, insulin secretagogues, DPP4 inhibitors, SGLT2 inhibitors, acarbose, TZDs, GLP1RAs, insulin) varied considerably, depending on whether these newly diagnosed T2DM patients initiated treatment earlier or later in the study window. Other than a different glycemic efficacy, the occurrence of first-year treatment failure may reflect either a specific medication intolerance, that often prevents patients maintaining their therapy, or therapeutic inertia, which is a non-negligible problem among diabetes care providers. Failing to control for these confounding variables, and their combined effects, may have distorted the associations, and should be regarded as an important study limitation.

**Response:**

Thank you for the insightful comment. As advised, we have added a controlled variable "enrollment time" in the multivariable logistic regression models to reduce the confounding effect. (Lines 131-132, Table 1, Table 2). Additionally, according to the reviewer's suggestion, we acknowledged the limitation in the "Discussion" section (Lines 246-249, 251-255).

2. Beside pharmacological treatment options, also glycemic targets have varied considerably during this wide study window, so that the definition of treatment failure used ("never achieving post-treatment HbA1c <8% at 3, 6, 9, or 12 months after initiating treatment during the first year", lines 94-95) appears misleading. It is true that before 2004, the American Diabetes Association (ADA) recommended that HbA1c not be allowed to exceed 8.0%. However, since 2004, the ADA dropped the traditional 8.0% "action threshold" in favor of a general recommendation to treat most patients to <7.0%, especially those newly diagnosed with T2DM and expected to get clinical advantage of metabolic legacy. Few exceptions to this general recommendation are represented by frail, older adults (> 65 years of age), in which less stringent HbA1c targets are recommended (8-8.5%) since late 2000s. I would suggest narrowing the study window to ensure more homogeneous criteria for treatment and outcomes of the study participants, redefining treatment failure, and excluding elderly patients with high Charlson comorbidity index scores from the primary analysis.

**Response:**

Thank you very much for helping us improve the quality of our manuscript.

We agree with this and have revised our models with narrowing the study window and redefining the definition of outcome (treatment failure).

(1) Redefining the outcome variable: "Treatment failure was defined as the HbA1c value >7% at the end of 1-year observation." (Lines 100-102)

(2) Excluding elderly patients (≥65 years) and those with high Charlson comorbidity index scores (≥5). (Lines 79-82)

Abstract, Results, Discussion and Conclusion were revised according to the revised models. (Lines 23, 25-26, 29-34, 38-39, 154-180, 187, 190, 193-205, 210-231, 268)

3. Although legitimate, backward stepwise regression modeling is particularly subjected to multicollinearity problems when the predictive variables are intercorrelated. The methods section should explain how multicollinearity was handled.

**Response:**

Thank you for reminding us this issue. As advised, we have revised and clarified this

point in Materials & Methods (Statistical analysis: Lines 139-142) and Results (Lines 172-174). Detailed data were presented in Supplementary Table.

| Variables | All variables | | Primary model | |
|---|---|---|---|---|
| | Tolerance | VIF | Tolerance | VIF |
| Age at onset | 0.707 | 1.414 | 0.737 | 1.357 |
| Gender | 0.734 | 1.362 | 0.777 | 1.287 |
| Level of education | 0.654 | 1.530 | 0.667 | 1.500 |
| Family history of DM | 0.913 | 1.095 | | |
| Current smoking | 0.803 | 1.246 | 0.810 | 1.235 |
| Physical activity | 0.904 | 1.106 | 0.913 | 1.095 |
| Medication adherence | 0.991 | 1.009 | 0.993 | 1.007 |
| Knowledge regarding GC | 0.657 | 1.523 | | |
| Willingness toward DSM | 0.949 | 1.054 | | |
| Perform SMBG | 0.781 | 1.280 | 0.774 | 1.292 |
| HbA1c at baseline | 0.905 | 1.105 | 0.931 | 1.074 |
| Mean BP | 0.958 | 1.044 | 0.964 | 1.037 |
| Total cholesterol | 0.653 | 1.532 | | |
| Triglycerides | 0.599 | 1.670 | 0.941 | 1.063 |
| HDL-C | 0.713 | 1.403 | | |
| Anti-diabetic Medication | 0.893 | 1.119 | 0.899 | 1.112 |
| Use of fibrates | 0.771 | 1.298 | | |
| Enrollment time | 0.597 | 1.675 | 0.760 | 1.316 |

## Response to Reviewer 2

Basic reporting    No comment

Experimental design     No comment

Validity of the findings    No comment

Comments for the author

This paper focuses on an interesting topic. However, there is a major concern that must be addressed.

The authors stated that "Participants with at least one of the four post-treatment HbA1c levels <8% were categorized as non-TF (reference group)". This critical choice must be strongly supported, otherwise it appears arbitrary and invalidates the study conclusions. Why didn't they choose only the HbA1c value at the end of the observation? In my opinion this value is more appropriate as cut-off between TF and

non-TF. In any case, please, show in table the HbA1c values at the four assessments.

**Response:**

We really appreciate your excellent comment.

(1) We agree with this and have revised our models with redefining the definition of outcome (treatment failure).

"Treatment failure was defined as the HbA1c value >7% at the end of 1-year observation." (Lines 100-102)

Abstract, Results, Discussion and Conclusion were revised according to the revised models. (Lines 23, 25-26, 29-34, 38-39, 154-180, 187, 190, 193-205, 210-231, 268)

(2)  The HbA1c values at the four assessments were shown in Table 1.